# Adaptive Budgerigar Optimization-based Obstacle Avoidance Path Planning for Unmanned Aerial Vehicle

Zhiqiang Li
*School of Aeronautics and Astronautics*
*University of Electronic Science and Technology of China*
Chengdu, China
zhiqiangli@uestc.edu.cn

Jingyu Liu
*School of Aeronautics and Astronautics*
*University of Electronic Science and Technology of China*
Chengdu, China
ljy202322100118@163.com

Mengji Shi
*School of Aeronautics and Astronautics*
*University of Electronic Science and Technology of China*
Chengdu, China
maangat@uestc.edu.cn

Boxian Lin*
*School of Aeronautics and Astronautics*
*University of Electronic Science and Technology of China*
Chengdu, China
linbx@uestc.edu.cn

Meng Li
*School of Aeronautics and Astronautics*
*University of Electronic Science and Technology of China*
Chengdu, China
mengli@uestc.edu.cn

Kaiyu Qin
*School of Aeronautics and Astronautics*
*University of Electronic Science and Technology of China*
Chengdu, China
kyqin@uestc.edu.cn

*Abstract*—This paper addresses the path planning problem of Unmanned Aerial Vehicle (UAV) operating in a three-dimensional environment populated with obstacles. An enhanced Adaptive Budgerigar Optimization (ABO) algorithm is designed to navigate the UAV efficiently, ensuring collision avoidance while maintaining high solution accuracy. The primary innovation of our approach involves modifying the iteration update formula of the original Parrot Optimization algorithm by incorporating an adaptive adjustment factor. This factor dynamically regulates the convergence rate and accuracy, thereby enabling the algorithm to escape local optima and achieve globally optimal paths effectively. Through comprehensive simulation experiments, we compare the performance of the ABO algorithm against traditional Particle Swarm Optimization (PSO) and the original Budgerigar Optimization algorithm. The results demonstrate the superior convergence speed and solution quality of the proposed algorithm, thereby validating its effectiveness and feasibility.

*Keywords—Path Planning; Adaptive Budgerigar Optimization (ABO); Obstacle Avoidance; Unmanned Aerial Vehicle*

## I. INTRODUCTION

In recent years, the continuous advancements in robotic automation technologies, including visual perception, path planning, and intelligent control, have led to the widespread adoption of Unmanned Aerial Vehicles (UAVs) across various fields. This is largely attributed to their compact size, cost-effectiveness, high mobility, and enhanced safety features. Among the key technologies enabling UAV autonomy and intelligence, path planning has emerged as a primary research focus [1-3]. Path planning for UAVs involves designing an optimal flight path that adheres to specific UAV and terrain constraints while avoiding obstacles, no-fly zones, and potential threats along the route. The efficiency and safety of UAV mission execution are directly influenced by the effectiveness of path planning. By optimizing the flight path, UAVs can achieve mission objectives in the shortest possible time, thereby enhancing overall operational efficiency.

Path planning methods are generally classified into three categories: traditional path planning algorithms, artificial intelligence algorithms, and swarm intelligence optimization algorithms. Traditional path planning algorithms[4] include methods such as the Artificial Potential Field (APF) algorithm [5] and the A* algorithm [6]. Artificial intelligence algorithms [7-9] primarily utilize techniques like neural networks, reinforcement learning, and other machine learning methods to address the UAV path planning problem. Swarm intelligence optimization algorithms [10] encompass approaches such as the Ant Colony Algorithm [11-12], Genetic Algorithm [13-14], and Particle Swarm Optimization (PSO) [15-17]. Unlike traditional path planning algorithms, which often suffer from high computational demands, limited applicability, and a propensity to converge to local optima, swarm intelligence algorithms are capable of global search, enabling the identification of optimal paths more effectively. Consequently, swarm intelligence algorithms are widely employed in UAV path planning.

However, it is important to note that current swarm intelligence algorithms still face several challenges, including limited optimization capabilities and high computational resource consumption. Additionally, many path planning

algorithms are tested in overly simplified scenarios, which may not adequately demonstrate their feasibility in more complex environments. To address these issues, researchers have developed the Parrot Optimization (PO) algorithm [18]. While this algorithm exhibits strong optimization performance, it continues to suffer from problems such as convergence to local optima and significant computational resource demands.

Based on the above analysis, this article conducted in-depth research on UAV path planning under obstacle constraints. The main innovative points are summarized as follows:

(1) A novel path planning scheme based on the Adaptive Budgerigar Optimization (ABO) algorithm is designed. This scheme enables effective obstacle avoidance in a three-dimensional environment while maintaining high solution accuracy. The UAV navigation efficiency and safety in complex environments are enhanced.

(2) The traditional PO algorithm in [18] is improved by introducing a unique adaptive adjustment factor. This factor dynamically regulates the convergence rate and accuracy, allowing for more effective escape from local optima and achievement of globally optimal paths. The enhanced algorithm demonstrates superior convergence speed and solution quality compared to the PO algorithm, particularly in scenarios requiring high-precision path planning.

The remainder of this paper is structured as follows: Section 2 presents the modeling of the UAV path planning problem in a three-dimensional environment. Section 3 introduces the ABO algorithm. Section 4 compares the proposed algorithm with the traditional PO and PSO algorithm, demonstrating its superiority and determining the optimal parameters through comparative experiments. Finally, Section 5 concludes the paper with a summary of the key findings.

## II. PATH PLANNING MODEL CONSTRUCTION

This section models the UAV mission planning problem and outlines the construction of randomly generated three-dimensional (3D) maps. Additionally, it details the flight range constraints and terrain limitations [19] that the UAV must adhere to during the path planning process.

### A. The model of task space

In this paper, we address the UAV path planning problem within a mountainous environment. To enhance the randomness and demonstrate the adaptability of the algorithm, the centers of the peaks are generated randomly. The heights of the mountains in this environment are modeled using an exponential function, and the corresponding mathematical model is represented as follows:

$$Z(x, y) = \sum_{a=1}^{n} h_t \exp[-(\frac{x - x_a}{x_{sa}})^2 - (\frac{y - y_a}{y_{sa}})^2], \quad (1)$$

where parameter $n$ represents the number of peaks; $x_a$ and $y_a$ are the center coordinates of the $a_{th}$ peak; $h_a$ is a geographic parameter, which controls the height degree; $x_{sa}$

and $y_{sa}$ are the $a_{th}$ peak's attenuation along the x-axis and y-axis directions, which control the slope of the peaks. The environment model is shown in Figure 1.

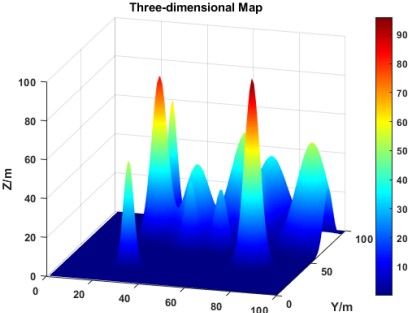

Fig. 1. Three-dimensional Map

### B. UAV path representation

The flight path of a UAV is characterized by an ordered set of point coordinates, each interconnected spatial coordinate point is associated using a cubic B-spline smoothing curve. Assume that the set of sequence points of the UAV path is $\{S, I_1, I_2, I_3 \cdots, I_{n-1}, T\}$. The group of node sequences are composed of $N+1$ nodes; S and T represent the starting and the target points of the UAV. The three-dimensional representation of starting and target point is $S = (x_0, y_0, z_0)$ and $T = (x_n, y_n, z_n)$, for the intermediate nodes in the path, we use $I_a = (x_a, y_a, z_a)(a = 1, 2, \cdots, n-1)$ to describe.

### C. Constraints and target function

The constraints are designed to ensure the planning of feasible flight paths. In the context of single UAV path planning, two primary constraints are introduced: terrain and environment.

Firstly, to prevent terrain collision during UAV flight, the flight altitude of UAV must consistently exceed the terrain altitude. Accordingly, the terrain constraints are formulated as follows:

$$Z_a > Z(x_a, y_a); \quad a = 1, 2, \cdots, n, \quad (2)$$

where $Z(x_a, y_a)$ is a terrain function, which can represent the height of the terrain at the position $(x_a, y_a)$.

Secondly, to optimize path planning and minimize costs during UAV flight, the UAV is restricted to operate within a designated area. The environmental constraints are therefore defined as follows:

$$\begin{cases} 0 \le x_a \le x_{max} \\ 0 \le y_a \le y_{max} \ ; \quad a = 1, 2, \cdots, n. \\ 0 \le z_a \le z_{max} \end{cases} \quad (3)$$

For the flyable path, the UAV's voyage obstacles, and boundary constraints must be collectively considered, enabling the generalization of the UAV's integrated cost function as follows:

$$\min(C) = \min(V_c + T_c + B_c), \qquad (4)$$

where $V_C$ represents voyage cost; $T_C$ represents terrain cost; and $B_C$ represents boundary cost.

The voyage cost $V_C$ essentially takes into account the distance flown by the UAV which is directly proportional to the distance $D$. It can be expressed as the total distance of the UAV from the starting point to the target point. If the total trajectory is composed of $n$ waypoints, the voyage cost can be represented as

$$V_C = \sum_{a=1}^{n-1} D_a. \qquad (5)$$

The terrain cost $V_c$ primarily accounts for the threat posed by mountain peaks along the UAV's flight path. The presence of this terrain cost significantly increases the expense of traversing mountainous obstacles, thereby optimizing the UAV's path to avoid such obstacles. It is expressed by the following mathematical formula:

$$\begin{cases} T_{c_0} = 0 \\ T_c = \sum_{a=1}^{n-1} T_{c_a} \\ T_{c_a} = \begin{cases} P, & z_a < z(x_a, y_a) \\ 0, & \text{else} \end{cases} \end{cases} , \qquad (6)$$

where the parameter $P$ represents the penalty for collisions with obstacles.

The boundary cost $B_c$ is primarily considered to ensure that the UAV operates within the designated spatial region during the flight, and is expressed as follows:

$$\begin{cases} B_{c_0} = 0 \\ B_c = \sum_{a=1}^{n-1} B_{c_a} ; \\ B_{c_a} = \begin{cases} P, & x_a \notin [0, x_{\max}] \text{ or } y_a \notin [0, y_{\max}] \text{ or } z_a \notin [0, z_{\max}] \\ 0, & \text{else} \end{cases} \end{cases} . \qquad (7)$$

Similarly, the parameter $P$ represents the penalty for increasing the cost if the UAV exceeds its flight range.

## III. ADAPTIVE BUDGERIGAR OPTIMIZATION ALGORITHM

Similar to other swarm intelligence optimization algorithms, the ABO algorithm is inspired by the natural behaviors of budgerigars. Previous studies have identified four distinct behavioral traits in budgerigars: searching, staying, communicating, and escaping. These behaviors serve as the foundational inspiration for the design of the ABO algorithm.

### A. Searching behavior

During searching behavior in ABO, the budgerigars determine the approximate location of the food by observation,

and then fly towards the estimated location. The population of budgerigar is assumed to be $N$ and the maximum number of iterations of the algorithm is $iter$, the behavior can be described by the following equation:

$$P_i^{t+1} = (P_i^t - P_{best}) \cdot Levy(n) + rand(0,1) \cdot (\alpha - \frac{t}{iter})^{\frac{(\alpha+1)t}{iter}} \cdot P_{mean}^t. \qquad (8)$$

In the equation (8), $P_i^t$ represents the current position, $P_i^{t+1}$ represents the position of the next momentary update. $P_{mean}^t$ represents the average position of all individuals in the current population. $P_{best}$ represents the optimal position searched so far. $rand(0,1)$ represents a random number generated from 0 and 1. $Levy(n)$ represents Levy flight strategy, where n represents the number of intermediate points selected throughout the path planning process. $t$ denotes the number of iterations so far.

For the parameter $\alpha$ in equation (8), is a dependency coefficient. The dependency coefficient is used to regulate the degree of dependency of an individual on the global search. By adjusting the size of the dependency factor, we can change the search weights of individuals globally and locally, which in turn gives better results.

The average position of all individuals can be calculated using the following equation：

$$P_{mean}^t = \frac{1}{N} \sum_{k=1}^{N} P_k^t. \qquad (9)$$

The Levy flight strategy can be represented by the following equation:

$$\begin{cases} Levy(n) = \frac{\mu \cdot \sigma}{|v|^{\frac{1}{\gamma}}} \\ \mu \sim N(0, n) \\ v \sim N(0, n) \\ \sigma = (\frac{\Gamma(1+\gamma) \cdot \sin(\frac{\pi\gamma}{2})}{\Gamma(\frac{1+\gamma}{2}) \cdot \gamma \cdot 2^{\frac{1+\gamma}{2}}})^{\gamma+1} \end{cases} . \qquad (10)$$

### B. Staying behavior

The budgerigar is highly social, it may randomly explore a certain location and stay there for a while based on existing experience, therefore, its behavior can be expressed by the following equation:

$$P_i^{t+1} = P_i^t + P_{best} \cdot Levy(n) + rand(0,1) \cdot ones(1,n), \qquad (11)$$

where $ones(1,n)$ represents the all-1 vector of dimension n. $P_{best} \cdot Levy(n)$ represents the behavior of its exploration and $rand(0,1) \cdot ones(1,n)$ represents the process of staying at a random location.

## C. Communicating behavior

As a natural social animal, budgerigar's behavior is inextricably linked to communication in a group. There are two types of budgerigar communication: flying back into the group and communicating with a single companion. We assume in the ABO algorithm that the two modes of communication have the same probability of occurring and consider the average position of all individuals in the population $P_{mean}^t$ to be the center of the population, we can express these two modes in the following two equation:

$$P_i^{t+1} = \begin{cases} 0.2 \cdot rand(0,1) \cdot (\beta - \dfrac{t}{iter}) \cdot (P_i^t - P_{mean}^t), & k \le 0.5 \\ 0.2 \cdot rand(0,1) \cdot exp(-\dfrac{t}{rand(0,1) \cdot iter}) \cdot P_i^t, & k > 0.5 \end{cases} \quad (12)$$

where $k$ in the equation is a random number from 0 to 1, which is used to randomly select which communication behavior occurs. $0.2 \cdot rand(0,1) \cdot (\beta - \dfrac{t}{iter}) \cdot (P_i^t - P_{mean}^t)$ represents the process of individuals flying back to the center of the population for communication and $0.2 \cdot rand(0,1) \cdot exp(-\dfrac{t}{rand(0,1) \cdot iter}) \cdot P_i^t$ denotes the process by which an individual communicates directly with other individuals. $\beta$, like $\alpha$ in equation (9), represents the dependency coefficient, which can be adjusted artificially.

## D. Escaping behavior

As a bird, Budgerigars has a natural fear of unfamiliar environments. When budgerigar is confronted with sudden dangers and situations, it will actively engage in evasion and return to a position it is familiar with and safe. Based on this behavior, we get the following equation:

$$P_i^{t+1} = P_i^t + rand(0,1) \cdot cos(0.5\pi \cdot \dfrac{t}{iter}) \cdot (P_{best} - P_i^t)$$
$$- cos(rand(0,1) \cdot \pi) \cdot (\dfrac{t}{iter})^{\frac{2}{iter}} \cdot (P_i^t - P_{best}), \quad (13)$$

where $rand(0,1) \cdot cos(0.5\pi \cdot \dfrac{t}{iter}) \cdot (P_{best} - P_i^t)$ represents the process of flying back to a safe position, and $cos(rand(0,1) \cdot \pi) \cdot (\dfrac{t}{iter})^{\frac{2}{iter}} \cdot (P_i^t - P_{best})$ represents the process of escaping a dangerous and unfamiliar environment.

## E. Explanation of the ABO algorithm

The optimization formula presented above is derived from the four behavioral traits of budgerigars. In the ABO algorithm, the behavior that an individual performs is determined by an identifier, known as the flag, which takes a positive integer value between 1 and 4, with each value having an equal probability of occurrence. Unlike traditional meta-heuristic algorithms, the ABO algorithm does not differentiate between the exploration and exploitation phases. Instead, individuals randomly select one of the four behaviors,

resulting in a straightforward algorithm with strong optimization capabilities.

## F. Flowchart of the ABO algorithm

The path planning process for the UAV using the ABO algorithm can be outlined in the following steps. First, the algorithm begins with the initialization of parameters, where the values of the hyperparameters are input. Next, the positions of the population are initialized, and the current value of the fitness function is calculated. Subsequently, the four behaviors of budgerigars are employed to update the positions, with the selection of behaviors determined by the random parameter flag. These behaviors optimize an individual's position by utilizing information such as the current optimal position and the average position of the population. The optimization process continues iteratively until the predefined stopping conditions are satisfied. The specific flowchart of the ABO algorithm is presented below:

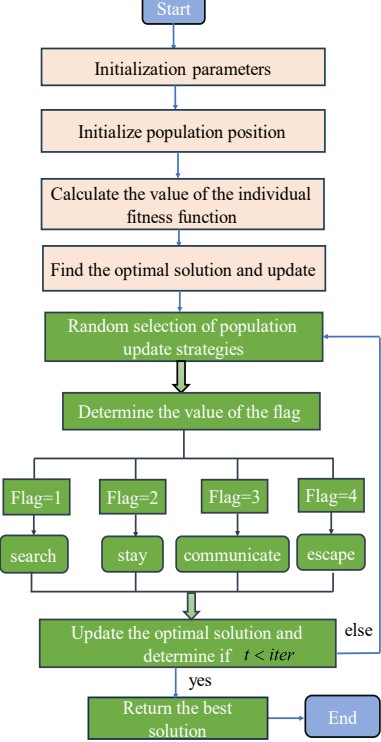

Fig. 2. the flowchart of the ABO algorithm

## IV. SIMULATION EXPERIMENT AND ANALYSIS

To verify the performance and advantage of the ABO algorithm in solving the UAV path planning problem, this paper takes the MATLAB environment for experimental simulation. Meanwhile, we compare the ABO algorithm with the traditional PSO algorithm. The superiority of the ABO algorithm can be seen from the results. Also to get the influence of hyperparameters $\alpha$ and $\beta$ on the path planning results, several sets of different values of $\alpha$ and $\beta$ are taken for comparison experiments to get the optimal parameter values.

## A. Simulation experiment based on ABO algorithm

In this part, we aim to demonstrate that the ABO algorithm is suitable for solving the UAV path planning problem. To achieve this, task scenarios are randomly generated to validate the algorithm's generalization capability. Additionally, the optimization performance of the ABO algorithm is assessed through comparative experiments with the PSO algorithm.

Simulation parameter settings: the number of intermediate points is three, the number of populations is set to 200, the number of iterations is set to 100, the coordinates of the starting point is [1,1,1], the coordinates of the target point is [85,80,40], and the three-dimensional coordinates are limited to [100,100,100]. The values of the adaptive parameters $\alpha$ and $\beta$ are both chosen to be 1. The number of peaks n is taken to be 25.In the environment of MATLAB 2022a, based on the ABO and PSO algorithm for UAV path planning we can get the following results:

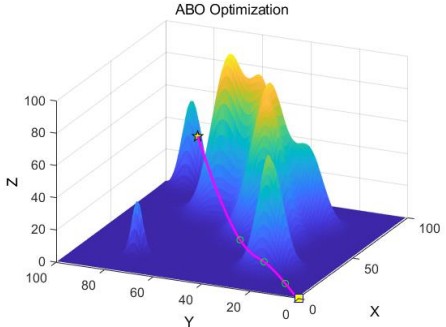

Fig. 3. The ABO algorithm path planning results

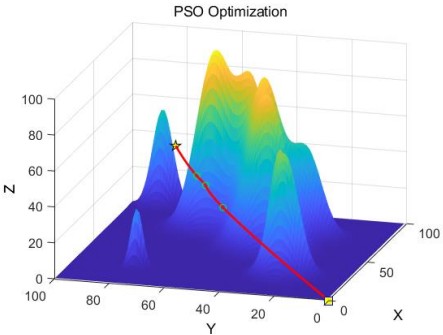

Fig. 4. The PSO algorithm path planning result

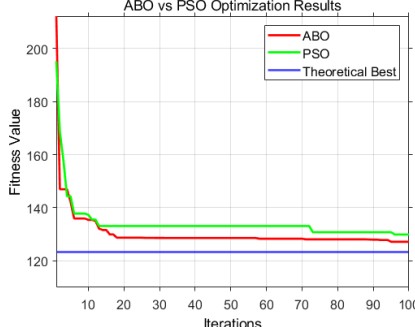

Fig. 5. Convergence curve by ABO and PSO algorithm

Figure 3 illustrates the UAV path planning results obtained using the ABO algorithm, while Figure 4 presents the results generated by the traditional PSO algorithm. Figure 5 displays the convergence curves for both the ABO and PSO algorithms, where the theoretical optimum represents the straight-line distance between the starting point and the target point. The simulation experimental results demonstrate that the ABO algorithm is a viable solution for the UAV path planning problem. Furthermore, the convergence curves in Figure 5 indicate that the optimization performance of the ABO algorithm surpasses that of the traditional PSO algorithm.

## B. Simulation experiments on adaptive parameter selection of the ABO algorithm

The previous simulation results have demonstrated the feasibility of the ABO algorithm in solving the UAV path planning problem. The next step involves conducting multiple sets of comparison experiments to determine the optimal adaptive parameters. Given that the maps used earlier were randomized, different trials could produce varying maps, potentially impacting the accuracy of the results. To address this, a fixed map will be used in the subsequent experiments to ensure consistency and accuracy.

The other parameters of the experiment are kept constant and only the values of the adaptive parameters $\alpha$ and $\beta$ are changed. Selecting 4 different sets of values of $\alpha$ and $\beta$, we can get the following results.

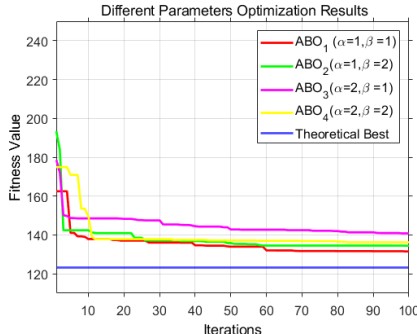

Fig. 6. Optimization results for different parameters

Figure 6 illustrates the optimization results of selecting four different sets of values of $\alpha$ and $\beta$. The red curve in Figure 6 is the case $\alpha = \beta =1$; the green curve is the case $\alpha =1$, $\beta =2$; the pink curve is the case $\alpha =2$, $\beta =1$ and the yellow curve is the case $\alpha = \beta =2$. From the results, we can see that with enough iterations, it is still the case of $\alpha = \beta =1$ that is optimized optimal.

## V. CONCLUSION

This paper has addressed the UAV path planning problem in three-dimensional environments with obstacles. An enhanced ABO algorithm, which efficiently navigates the UAV while ensuring collision avoidance and high solution accuracy is proposed. By incorporating an adaptive adjustment factor into the iteration update formula, the algorithm effectively escapes local optima and achieves globally optimal paths. Simulation results demonstrated the ABO algorithm's superior convergence speed and solution

quality compared to traditional methods, validating its effectiveness. Future research will focus on multi-UAV cooperative path planning to further optimize coordination and efficiency.

## ACKNOWLEDGMENT

This work was supported by the Natural Science Foundation of Sichuan Province (2022NSFSC0037), the Sichuan Science and Technology Programs (2022JDR0107, 2021YFG0130, MZGC20230069), the Fundamental Research Funds for the Central Universities (ZYGX2020J020), the Wuhu Science and Technology Plan Project (2022yf23) (Corresponding author: Boxian Lin).

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
