# OpenReview forum: "Adaptive Budgerigar Optimization-based Obstacle Avoidance Path Planning for Unmanned Aerial Vehicle"
_IEEE.org/ICIST/2024/Conference — IEEE ICIST 2024 Conference Submission_

### Official Review · Reviewer_JuWi · 2024-08-21
**The paper is logically clear, the simulation results are credible, and it is recommended for publication.**

**Rating:** 7
**Confidence:** 3

**Review:**

This paper presents a noteworthy contribution to the field of UAV path planning by introducing an enhanced Adaptive Budgerigar Optimization (ABO) algorithm tailored for navigating three-dimensional environments with obstacles. The core innovation lies in the modification of the iteration update formula from the original Parrot Optimization algorithm, incorporating an adaptive adjustment factor that dynamically controls convergence rate and accuracy. This strategic adjustment significantly improves the algorithm's ability to escape local optima and find globally optimal paths. The reviewer has the following questions to discuss with the authors:

1. How does the adaptive adjustment factor in the ABO algorithm specifically regulate the convergence rate and accuracy? What criteria are used to determine when and how the factor is adjusted during the optimization process?

2. The paper mentions that ABO outperforms traditional Particle Swarm Optimization (PSO) in convergence speed and solution quality. What are the key differences in the performance of these algorithms, and are there particular situations where PSO might still be preferable?

3. How does the computational complexity of the ABO algorithm compare to that of the original Budgerigar Optimization and PSO algorithms? Is the improved performance achieved at the cost of higher computational demands?

---

### Official Review · Reviewer_mhGR · 2024-08-21
**This manuscript has a certain degree of innovation and clear simulation figures. It is recommended to accept this paper for publication in IEEE ICIST 2024.**

**Rating:** 8
**Confidence:** 4

**Review:**

This paper addresses the path planning problem of Unmanned Aerial Vehicle (UAV) operating in a three dimensional environment populated with obstacles. This manuscript has a certain degree of innovation and clear simulation figures. It is recommended to accept this paper for publication in IEEE ICIST 2024. However, the following suggestions need careful consideration to further improve the quality of the paper：
1.The ABO algorithm has demonstrated its effectiveness in single-UAV path planning. To further enhance its practical applicability, extending the algorithm to support multi-UAV cooperative path planning would be highly valuable. By incorporating mechanisms for coordinating and optimizing the paths of multiple UAVs simultaneously, the proposed approach could achieve even higher operational efficiency and robustness in real-world missions involving teams of UAVs.
2.While the ABO algorithm effectively navigates UAVs around static obstacles, integrating real-time obstacle detection capabilities using sensors such as LiDAR, RADAR, or cameras could significantly enhance the system's adaptability to dynamic environments. This would allow the UAVs to respond to unforeseen obstacles encountered during flight, thereby improving overall safety and reliability.
3.The current ABO algorithm focuses on minimizing the overall path length and avoiding obstacles. To further optimize the performance of UAVs, incorporating energy-aware path planning into the algorithm would be beneficial. By considering factors like altitude, speed, and wind conditions into the cost function, the algorithm could generate paths that minimize energy consumption, extending the operational range and endurance of UAVs in real-world missions.

---

### Official Review · Reviewer_Ncrs · 2024-08-21
**Accept**

**Rating:** 7
**Confidence:** 5

**Review:**

The paper proposes an innovative enhancement to the Budgerigar Optimization algorithm, introducing an adaptive adjustment factor to tackle the complex path planning problem for UAVs in 3D environments with obstacles. This work is significant as it aims to improve the efficiency and accuracy of path planning algorithms.However,there are some suggestions :
1.Comparing ABO against a wider range of optimization algorithms, especially those specifically designed for path planning, would provide a more comprehensive evaluation.
2.Improve the clarity and conciseness of the writing, particularly in the introduction and related work sections.
3.It is better to  discuss the impact of parameter settings.

---

### Decision · Program_Chairs · 2024-09-08

Accept (Oral)